# Comparison of Clinical Effectiveness of Deslorelin Acetate and Osaterone Acetate in Dogs with Benign Prostatic Hyperplasia

**DOI:** 10.3390/ani10101936

**Published:** 2020-10-21

**Authors:** Wojciech Niżański, Małgorzata Ochota, Christelle Fontaine, Joanna Pasikowska

**Affiliations:** 1Department of Reproduction and Clinic of Farm Animals, Faculty of Veterinary Medicine, Wroclaw University of Environmental and Life Sciences, pl. Grunwaldzki 49, 50-366 Wroclaw, Poland; malgorzata.ochota@upwr.edu.pl; 2Virbac Group, Global Marketing and Business Optimization Department, Companion Animals Veterinary Exclusive Ranges Section, 13ème rue LID, 06511 Carros, France; christelle.speiser-fontaine@virbac.com; 3Veterinary Clinic, B. Krzywoustego Str. 105/22, 51-166 Wroclaw, Poland; j.h.pasikowska@gmail.com

**Keywords:** dog, BPH, treatment, osaterone acetate, deslorelin acetate

## Abstract

**Simple Summary:**

The article compares the treatment efficacy and adverse effects of two drugs used for benign prostate hyperplasia (BPH) therapy in dogs: Ypozane^TM^ (osaterone acetate) and Suprelorin^TM^ (deslorelin acetate). Ypozane^TM^ is a registered medication for this condition in dogs, whereas Suprelorin^TM^ is registered for pharmacological castration in dogs. The clinical trial proved both drugs to be safe and effective in reducing BPH-related symptoms in dogs, and the noted adverse effects were only mild—mostly weight gain. With osaterone acetate the clinical improvement was noted sooner (from day 7 onwards) than with Suprelorin^TM^ (from day 21 onwards), but it lasted shorter up to 24 weeks, while in the Suprelorin^TM^ group, clinical effect remained stable until the end of the study (36 weeks). Both medications can be recommended for treatment of symptoms related to BPH in dogs, as none of the drugs had serious influence on the general health status and both provided substantial clinical improvement.

**Abstract:**

This article presents the results of a randomized clinical trial, designed to compare the efficacy and therapeutic profiles of Ypozane^TM^ (osaterone acetate—OA) or Suprelorin^TM^ (deslorelin acetate—DA) in male dogs with clinical signs of benign prostate hyperplasia (BPH). Forty-five intact male dogs were used in the study. The Group I (negative control) included 10 healthy dogs, the Group II (positive control) included 10 dogs with confirmed BPH and no treatment, whereas Group III and IV consisted of dogs with BPH and treated either with DA (15 dogs) or OA (10 dogs). The clinical response, testosterone and estradiol levels, hematology, biochemistry, and adverse effects incidence were evaluated. Both OA and DA proved to be effective for BPH treatment in dogs, as they allowed for the clinical remission in all treated dogs. The complete alleviation of BPH symptoms was noticed sooner with the use of OA (in 80% of dogs from day 7) compared to DA (in 40% of dogs within the first 21 days). The recurrence of clinical signs related to BPH was observed from week 24 in dogs treated with OA, whereas no relapse was noticed in dogs treated with DA at the end of the 36 weeks of the observation period. In 5 dogs (33%) treated with DA, a flare-up effect (increase in the clinical signs associated with BPH) was noticed on day 7. Despite individual differences in the clinical action, both medications were effective and safe options for the treatment of symptoms related to BPH in dogs.

## 1. Introduction

Among others, the benign prostatic hyperplasia (BPH) is one of the most common age-related conditions seen in male, intact dogs. BPH is the androgen-dependent multiplication of the prostatic cell number (hyperplasia) with the subsequent increase in the cell size (hypertrophy). In dogs, the main circulating androgen is testosterone produced by Leydig cells of the testes and converted by type II 5α-reductase into dihydrotestosterone (DHT) in prostate, seminal vesicles, epididymies, skin, liver, and brain. Since dogs (apart from some primates) are the only nonhuman species diagnosed with spontaneous, age-related benign prostatic hyperplasia, canine BPH cases might become a valuable model to study this problem in humans [1,2].

Around 80% of sexually intact dogs older than 5 years and 95% dogs older than 9 years show gross or microscopic changes related to BPH [3,4,5]. Despite that, at the beginning, the condition is often unnoticed, and the reason for consultation would be the reduced quality of life or accompanying symptoms, i.e., locomotor disorders, hyperthermia, unexplained deterioration of body condition, or GI symptoms (constipation, tenesmus, or intermittent diarrhea). Only later, with the progressive enlargement of the prostatic gland, there would be observed a sanguineous discharge from the prepuce or urethra, hematuria, and hemospermia, ribbon-like appearance of feces and sometimes rectal obstruction [6]. As BPH could impair fertility due to decreased libido or blood detection in the ejaculate, in stud dogs the prostatic disorders are often diagnosed earlier, before the late symptomatic stage occurs [7,8].

The treatment method of choice in non-breeding dogs is a surgical or pharmacological castration. Testes are the main source of testosterone, so its removal and the subsequent decrease in testosterone levels lead to the shrinkage of the gland, which around 9 weeks after castration, becomes no more than 25–30% of its initial size [7]. Unfortunately, the surgical castration is not always feasible, e.g., due to a higher anesthetic risk related to age or concurrent disorders [9], or arising ethical concerns make surgical castration a questionable, or even inacceptable, method of treatment. In stud dogs, maintaining the fertility is the key factor, whereas in working dogs, testosterone is essential for better performance. In such cases, pharmacological therapy is to be considered. Various pharmacological agents were tried for BPH from estrogens, antiandrogens, 5α-reductase inhibitors, progestogens to experimental therapies utilizing micronized endogenous fatty acid amides and resveratrol precursors [2,5,10,11]. However, only osaterone acetate has been registered for BPH treatment in dogs. The active substance of osaterone acetate is the steroidal antiandrogen, which treats BPH by reducing the uptake of androgens in prostatic tissue and inhibiting the action of 5α-reductase. Osaterone acetate (OA, Ypozane^TM^ Virbac, Carros, France) is marketed as oral tablets in four different strengths depending on the dog’s body weight, relieving clinical symptoms for five months after a 7-day treatment course. Its use has been widely studied and proved to be very effective for BPH therapy. Moreover, stud dogs under the OA treatment remain fertile and may still be used for mating [2,9,12,13,14].

Recently, another drug has been suggested for BPH treatment in dogs, the GnRH agonist deslorelin acetate (DA, Suprelorin^TM^, Virbac, Carros, France). Deslorelin acetate is GnRH superagonist, which after the initial rise in pituitary gland hormones, leads to desensitization of the receptors responsible for the gonadotrophin release, followed by the dose- and time-dependent depletion in LH and FSH, which finally markedly decreases the production of androgens. For dogs, it is manufactured as a subcutaneous implant distributed in two sizes and induces infertility for at least 6–12 months. Apart from stopping the sperm production and reducing libido and sexual behavior, it also affects the prostate gland, decreasing its size which is similar to the effect seen from surgical castration. Unfortunately, there is only limited data available on DA efficacy for BPH treatment in dogs [15,16,17], and up to now, no comparative studies have been published concerning the biological effects of DA in dogs.

This article presents the comparison of the treatment efficacy of osaterone acetate or deslorelin acetate in male dogs suffering from benign prostate hyperplasia.

## 2. Materials and Methods

The trial has been approved by the II Local Ethics Committee for Animal Experiments of the University of Environmental and Life Sciences in Wroclaw (No 36/2014).

### 2.1. Animal Selection

The 45 intact male dogs aged more than 5 years of different breeds (Table 1) were enrolled for the study. Dogs were divided into four groups (two control and two treated groups).

Control groups: Group I (negative control group) comprised 10 healthy dogs aged 5–10 years (mean 6.5; SD ± 2) and weighing 8.7–23.5 kg (mean 17.08; SD ± 6.21). Group II (positive control group) included 10 dogs, aged 5–15 years (mean 9.5; SD ± 3.5) and weighing 2.9–44 kg (mean 14.12; SD ± 12.17), diagnosed with BPH, with no treatment administered.

Treated groups: Group III (deslorelin acetate treatment group) included 15 dogs aged 6–15 years (mean 9.47; SD ± 2.13), weighing 7.6–46.5 kg (mean 27.6; SD ± 14.65), diagnosed with BPH and treated with deslorelin acetate. Group IV (osaterone acetate treatment group) comprised 10 dogs, aged 5–10 years (mean 7.6; SD ± 1.96) and weighing 4.1–50.5 kg (mean 26.92; SD ± 15.25), diagnosed with BPH and treated with osaterone acetate.

### 2.2. Indications for Inclusion into Control Group II and Treated Groups III and IV

Dogs showing one or more of the following symptoms: sanguineous discharge from the prepuce or urethra, hematuria and hemospermia, tenesmus, or straining during defecation were qualified for further assignment. Dogs were allocated to the control (II) or the treated groups (III and IV) based on the initial examination, which included: history and interview with owner; clinical examination and weighting of the patient; rectal palpation of the prostate; complete blood count, serum biochemical analysis, and serum hormonal analysis; FNA (fine needle aspirate) of the prostate (US-guided FNA) in order to confirm BPH by cytological criteria [18,19].

### 2.3. BPH Diagnosis

In dogs allocated into control (II) and treated (III, IV) groups the confirmation of BPH was based on: reported clinical signs, symmetrical enlargement of the prostate on rectal palpation, and the results of the prostate biopsy. To compare the severity of exact symptom related to BPH, the objective score system for BPH signs grading, proposed by Zambelli [20], was used. In brief, dogs suspected with BPH were assessed for the presence, severity, and duration of exact symptoms. Each symptom was evaluated and graded as 1 = absent, 2 = light and/or short duration, 3 = moderate severity and/or middle duration, and 4 = severe and/or long lasting (Table 2).

### 2.4. Drugs and Treatment Schedule

Group III on D0, after performing the prostate biopsy, received the 4.7 mg deslorelin acetate implant (Suprelorin^TM^ Virbac, Carros, France), inserted subcutaneously on the dogs’ back.

Group IV on D0, after performing the prostate biopsy, were started on the osaterone acetate (Ypozane^TM^ Virbac, Carros, France) tablets, administered orally at a dose recommended by the manufacturer: 0.25–0.5 mg/kg every 24 h for 7 consecutive days.

### 2.5. Study Design

The trial was designed as monocentric, randomized controlled, clinical, and non-blinded study. The patients were allocated into one of the II–IV treatment groups and treated as described above (Group III and IV) or not treated and served as positive control group (Group II).

All the 45 treated dogs were examined on Day 0 (D0); Day 7 (D7); Day 14 (D14); Day 21 (D21) and Week + 8 (W8); Week + 12 (W12); Week + 16 (W16); Week + 20 (W20); and Week + 24 (W24). Three months after the trial finished (Week + 36, W36), i.e., after the manufacturer’s guaranteed duration of action, all dogs from experimental Groups III and IV underwent the follow-up examination to evaluate potential long-term effects of both drugs.

Each time the examination included: history and interview with owner; clinical examination and weighting of the patient; rectal palpation of the prostate; complete blood count, serum biochemical analysis, and serum hormonal analysis. On Day 0 in all dogs from Group II, III, and IV the US-guided FNA of the prostate was performed in order to confirm BPH by cytological criteria.

The results obtained in the treated Groups III and IV were compared to D0 values in corresponding groups; between both (III vs. IV) treated groups and to the results obtained in both control groups (I and II) (Figure 1).

### 2.6. Statistical Analysis

Statistical analysis was carried out using the STATISTICA 10.0 software (StatSoft, Kraków, Poland). The blood morphology, biochemistry, and hormonal assays obtained on D0 differed among all the investigated groups depending on the physiological status of the individual dogs included, hence the results were analyzed only with respect to the D0 values [17,21]. The values within treated groups were compared to D0, to each other, and to the values obtained in control and treated groups, using the paired Student’s *t*-test or Mann–Whitney’s U-test depending on distribution. A probability value at *p* = 0.05 was considered statistically significant.

## 3. Results

First, there were compared the results between Group I (negative control group) and Group II (positive control group), to identify any differences between healthy dogs and dogs with BPH. Secondly, results obtained in Group III and Group IV were compared to Group II, which served as a positive (untreated) control group, for both treatment groups (III and IV), to identify any differences between treated and not treated dogs with BPH. Lastly, the Groups III and IV were compared to each other to see any differences in the onset of clinical improvement, treatment efficiency, treatment duration, or observed adverse effects (Figure 2).

### 3.1. Clinical Response

At least two clinical signs typical for BPH were noted in 90% of dogs form Group II and in 100% of dogs in Group III and IV. Most often, it was hemospermia (100% of investigated semen samples); sanguineous discharge from the prepuce or urethra, noted in all dogs from Group III and IV and 93.3% dogs from Group II; or hematuria (Group II—40%, III—86.7%, IV—70% of investigated dogs). Straining during defecation, ribbon-like appearance of feces, or constipation were seen less often. Pain on rectal palpation was noted only in 20% and 10% of patients from Group III and IV, respectively. None of the investigated dogs showed symptoms of dysuria (Figure 2).

In the objective score system of BPH symptoms assessment [18], no statistical difference was noted among all the investigated groups during initial qualification. Patients from Group II obtained 22.8 points (± 6.34), from Group III—24 points (± 4.39), and from Group IV—23.4 points (± 5.4) (Figure 3).

Clinical symptoms were present in all dogs from Group II throughout the whole studied period with intermittent exacerbation of clinical signs in some of the dogs. In 40% (six dogs) of dogs from Group III, all symptoms subsided within the first 21 days after inserting the implant, whereas in the remaining 60% (9 dogs), symptoms subsided in week 8 (W + 8) after inserting the implant. In five dogs on D7, the flare-up effect was noted (aggravation of clinical signs, i.e., hematuria, sanguineous discharge from the prepuce, hemospermia, tenesmus, or constipation) and in two of these dogs, it was still present on D14 (these dogs have the most severe BPH symptoms from all the dogs that entered the trial). In Group IV, 80% of treated dogs showed no symptoms on D7 after starting the therapy, while the remaining 20% had no symptoms on D14. However, four dogs (40%) in Group IV showed signs of relapse, i.e., clinical signs recurrence in W + 24 of the study (Figure 4)

### 3.2. Hormone Assays and Blood Parameters

Both the serum 17β-estradiol and testosterone levels in all dogs from Group I and II were within normal limits and did not change significantly throughout the trial. In Group III and IV the serum testosterone and 17β-estradiol levels were normal at the beginning of the trial and decreased gradually during treatment. Testosterone: In Group III after a transient rise on D7 testosterone was decreasing to non-detectable values in 40% of dogs in W8 and in all dogs in W16. The significant reduction was observed on D14 in comparison to Group II and on D21 in comparison to D0, and the quickest decrease was noted between D7–D14. In Group IV, the lowest testosterone level was observed in W + 16, however no difference was noted comparing to the initial testosterone levels in this group and, compared to the control Group II, the significantly lower testosterone levels was detected in W + 5 (Figure 5). 17β-estradiol: in Group III, a significant reduction was noted in W + 12 of the therapy and was still decreasing until the end of the trial. In Group IV, after the initial increase comparing the initial value and Group II value (31.69 pg/mL ± 14.8 vs. 13.26 pg/mL ± 10.6; *p* = 0.008), thereafter the 17β-estradiol levels decreased and reached values similar to D0 (Figure 6).

Complete blood count and serum biochemical analysis. In all analyzed groups the hematological and biochemical parameters were within normal limits. Some differences were noted among individuals, however the mean values remained within normal limits (Table 3).

### 3.3. Treatment-Related Adverse Effect

In Group III and IV, we observed an increase in appetite (owners questionnaire) and weight gain. It started within the first 8 weeks of the treatment for both drugs and continued until the end of the study (W + 24). It was more prominent in Group III: 93.3% of dogs gained weight (1.84 kg on average), whereas in Group IV only 40% gained weight (0.86 kg on average). There were no change in body weight in dogs from Group I and II.

In two dogs from Group III, we observed the lower activity levels in W + 8 and in three dogs (in one on D7 and in two on D21) clinical symptoms of cystitis (incontinence, dysuria, or stranguria), which were resolved on symptomatic treatment (antibiotics and non-steroidal anti-inflammatory drugs) within one week. In Group IV, lower activity levels were also noted in two dogs (D14) and transient molting and dull hair coat were reported by owners in two dogs in W + 12 and W + 16, respectively. The above fur problems have settled quickly, as they were not mentioned again in the following questionnaires by the owners. In some patients from this group (IV), incidental gastro-intestinal symptoms or occasional polyuria and polydipsia were also seen (Figure 7).

In individual dogs form Group III the following clinical improvements were reported: in one dog, the reduction in anal glands’ adenomas and hair re-growing on supracaudal gland hyperplasia (stud tail), in a second dog with ongoing alopecia X, the noticeable hair growth, and in the third one, the visible reduction in perineal hernia sac.

In week 36 (3 months after the trial finished), dogs from Group III and IV were examined again. It is after the manufacturer’s guaranteed duration of action for both medications used in the study (DA and OA) to evaluate potential long-term effects of both drugs. During this evaluation, none of the dogs from Group III showed symptoms related to BPH. While 75% of examined dogs form Group IV had sanguineous discharge from the prepuce or urethra, hematuria, or hemospermia. The hematology and biochemistry were within reference limits in both groups. Similarly the 17β-estradiol levels were low in both groups (<9 pg/mL in Group III and IV), while the testosterone was lower in Group III (0.66 ng/mL) comparing to Group IV (1.73 ng/mL) in W + 36.

## 4. Discussion

The purpose of this clinical study was to compare the therapeutic efficacy of two medications used in BPH treatment for dogs: osaterone acetate (OA) and deslorelin acetate (DA). The study was designed as a randomized, non-blinded controlled clinical trial and is, to the best of the authors’ knowledge, the first publication comparing the therapeutic potential and safety of the OA and DA for the management of BPH in dogs.

### 4.1. Clinical Efficiency

Benign prostatic hyperplasia (BPH) is the most common prostatic disease diagnosed in dogs. It is a spontaneous and age-related condition occurring in older dogs. Initially, it is often asymptomatic or with very mild, unnoticed clinical signs. Serious clinical problems develop with time, after a considerable progress of the disease [3]. All dogs enrolled in the presented clinical trial had clinical signs of BPH, which significantly reduced the patient’s quality of life. Similar to other authors [3,20,22], in our study, we observed most often intermittent sanguineous discharge from prepuce or urethra without micturition, hemospermia, gross hematuria, and straining during defecation. The objective score system for the assessment of the severity of clinical symptoms [20] used in our trial guaranteed the optimal patient selection for each group and provided the highest possible coherence among the treated groups. Moreover, it also allowed to highlight the differences among groups during the trial.

In dogs treated with OA clinical signs fully improved during the first 14 days after starting the treatment, which is in agreement with already published data. Albouy et al. [13] observed BPH symptoms’ cessation between 2 and 3 week, whereas Tsustsui [9] within the first week of treatment. Although, with the use of OA clinical signs disappeared sooner than with DA, in four dogs, we observed the re-enlargement of the prostate gland together with the recurrence of the clinical signs in W + 24. Similar observations were reported by Albouy et al. [13]. He observed a complete remission of clinical signs throughout the six month trial period in most of the dogs, apart from 16%, which during the follow-up visits showed at some point the clinical relapse and the increase in the prostate volume. Nevertheless, it should be noted that both drugs, OA and DA, were clearly effective in clinical signs reduction. Dogs treated with OA responded quicker (W + 2) and without a flare-up effect, compared to DA treated dogs, where the improvement was noted later (W + 3 − W + 8), but lasted longer (up to W + 36). Osaterone acetate, as reported by Murakoshi et al. [23], causes a marked atrophy of the prostate glandular epithelium leading to the regression of BPH symptoms, but only for a limited period of time. On the other hand, Junaidi et al. [24] demonstrated that the microscopic tissue regression caused by DA implant was similar to that induced after surgical castration. Hence, it can be stated that clinically the pharmacological castration, obtained with the use of deslorelin implant, provides equal efficacy in the reduction in the BPH symptoms as the surgical castration.

The only published trials presenting the use of DA in dogs was performed in clinically asymptomatic dogs in which the prostate gland showed signs of enlargement only on US and histological examinations [17]. In this study, all dogs in treated groups have clinical symptoms typical for BPH. With the use of DA clinical symptoms disappeared in all treated dogs within 8 weeks after inserting the DA implant, which is similar to the results of Goericke-Pesch et al.’s study [21], with the use of another GnRH agonist-nafarelin implant. Additionally, we observed a worsening of the clinical signs of BPH, during the flare-up phase in one third of the investigated dogs treated with DA. The initial exacerbation of clinical symptoms is a typical finding reported with the use of many other receptor agonists. In case of GnRH agonist, it was suggested by Junaidi et al. [24,25] that DA leads to the LH and the subsequent testosterone surge, which directly affects the libido and sexual behavior of the patient. In such cases, it seems advisable to use DA implant in combination with androgen-receptor blockers during the first few days of the therapy, which was also recommended by Riesenbeck et al. [26].

### 4.2. Serum Testosterone and 17β-estradiol Levels

Our results were the same as those already published and confirmed that DA leads to a progressive atrophy of Leydig cells and testicular down regulation with the subsequent decrease in serum testosterone levels, which became undetectable around week 8 of the therapy. Since testicles are the only source of the testosterone in the body, the low serum testosterone levels directly indicate their function. [17,24,27,28]. On the contrary, the reports on the influence of OA on the serum testosterone levels are inconsistent. Murakoshi et al. [23] documented that OA has no effect on Leydig cells, thus the serum testosterone levels should remain stable during treatment. However, in the same paper, Murakoshi et al. [23] reported a transient decline in testosterone level during OA treatment. Same findings regarding testosterone drop were also reported by Tsutsui et al. [10,12]. Moreover, the serum levels of testosterone were lower in groups of dogs receiving higher OA doses (exceeding manufacturer recommendations, > 1 mg/kg BW) [9]. In our trial, we also observed a slight drop in testosterone levels, insignificant comparing to D0 values within the same group of investigated dogs, but in W + 8, significantly lower comparing to the both control group (Group I and II). Hence, although subtle, the antiandrogenic effect of OA cannot be fully excluded [23], but as found by other authors, it did not impair the process of spermatogenesis [2,9].

The already-published data show a substantial variability in plasma estradiol concentrations in an individual dog and groups of dogs, regardless the time of collection, sex, or hormonal status. The reason for this variability was related to the fact that estradiol can be produced by many tissues in the body [29]. Our results also confirmed a wide range of 17β-estradiol levels observed in the investigated dogs.

The present study reports for the first time the serum 17β-estradiol levels in dogs treated with deslorelin acetate implants. We noted that estrogen levels gradually decreased during the trial, reaching 1.56 pg/mL in W + 24, which is similar to the results presented with the use of another GnRH agonist—nafarelin [21,30]. The decrease in the serum estradiol levels observed in our study clearly demonstrates a downregulation of the hypothalamo-pituitary–gonadal axis caused by DA. However, the noted suppression of the estradiol production was not as distinct as for testosterone, which is most likely due to the various sources of estrogens in the body, whereas testosterone originates only from the testicles. On the other hand, dogs receiving OA showed a significant rise in the estradiol levels on D7 after starting the therapy. Considering the OA way of action, it could be suspected that the antiandrogenic properties of the osaterone acetate would block the testosterone receptors, leading to the rise in the free, circulating testosterone. However, both the other researchers and us failed to observe such rise in the serum testosterone levels during OA treatment [10,12]. It might be hypothesized that the circulating testosterone, unable to bind to its (blocked by OA) receptors, would be transformed into estradiol, resulting in a rise in serum of the latter one. Or, as suggested by Granner [31], free testosterone may competitively bind to sex hormone binding globulin (SHBG) replacing estradiol, hence its rise in serum. Normally, only a very small fraction of testosterone and estradiol circulate in the bloodstream as unbound particles; the majority is bound to serum albumin or SHBA, which inhibits the function of these hormones. Since, the relative binding affinity is higher for the testosterone than estrogens, it maybe the other explanation of the elevated levels of estrogens in dogs treated with OA.

### 4.3. General Blood Results

The obtained general blood tests results of dogs treated with OA and DA confirmed that both drugs are safe, neither affect liver or kidney function, nor bone marrow activity. The latter one is especially important as some hormonal drugs may lead to bone marrow suppression, erythropenia, and thrombocytopenia. The dogs in our trial were closely monitored up to week 36 of the treatment, and based on the obtained results, it can be stated that both drugs are a safe option for BPH treatment. Similarly to other reports using pharmacological [12] and surgical castration [32], the most common adverse effect seen in our study was the body weight increase (93% and 40% of the investigated dogs treated with DA and OA, respectively). Furthermore, in dogs receiving OA, we observed behavioral changes (dullness), transient digestive problems, and sometimes, polyuria and polydipsia. In all the investigated dogs the symptoms were mild to moderate proving the safety of both osaterone and deslorelin usage. Interestingly, in some of the investigated dogs, the noted side effects of DA implant were beneficial and improved the clinical state of sex hormone related disorders, i.e., alopecia X and perianal glands tumors, which appear to be hormone-related conditions [33].

## 5. Conclusions

Both osaterone acetate and deslorelin acetate were effective for BPH treatment in dogs. Both allowed for clinical remission and alleviated all connected to BPH symptoms. The clinical improvement was observed sooner with the use of OA but lasted longer with the DA treatment (at least 36 weeks). Apart from mild adverse effects (mostly weight gain) none of the drugs had a serious influence on the general health status. Both medications can be recommended for treatment of symptoms related to BPH. However, with DA, there might be a flare-up effect at the beginning of the therapy, while the clinical relapse with OA would be seen earlier from W + 24 onwards. Based on the obtained results, in some cases, it might be justified to consider the combined therapy with both medications. The OA to be given first and followed by DA 1–2 weeks later. Additional studies are needed to check whether the combined therapy using OA and DA would lead to a better clinical efficacy, reducing the adverse effects of both medications, i.e., exacerbation of clinical symptoms seen initially with DA and the limited time of clinical improvement seen with the use of OA.

## Figures and Tables

**Figure 1 animals-10-01936-f001:**
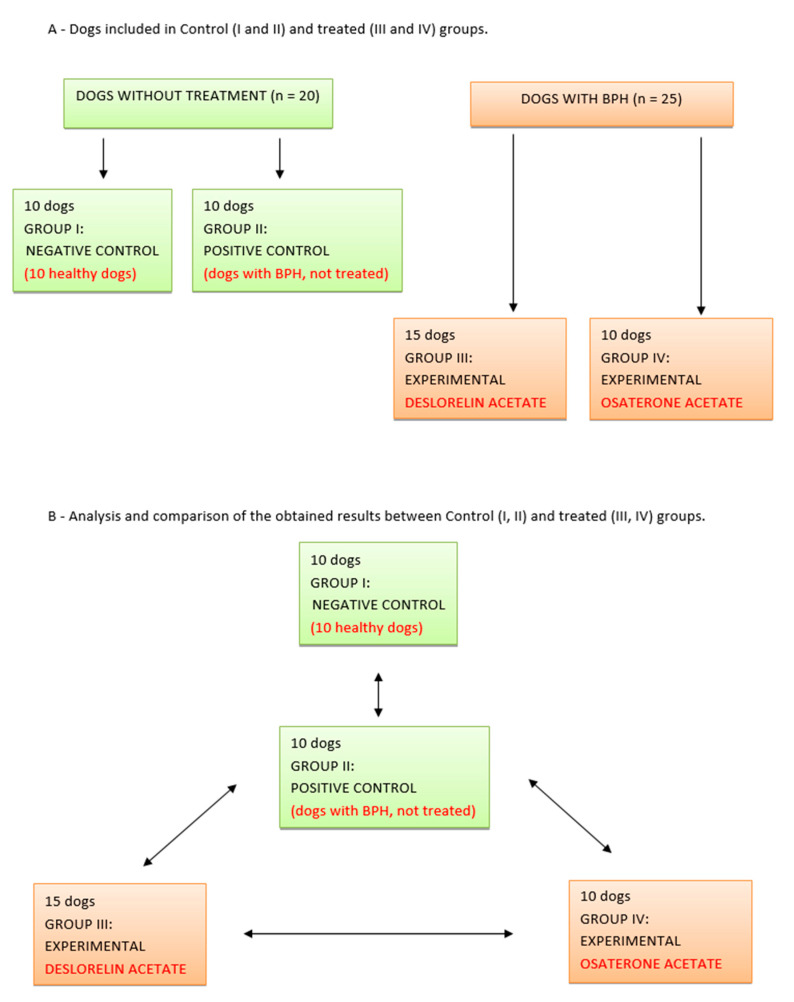
Study design (BPH—benign prostate hyperplasia).

**Figure 2 animals-10-01936-f002:**
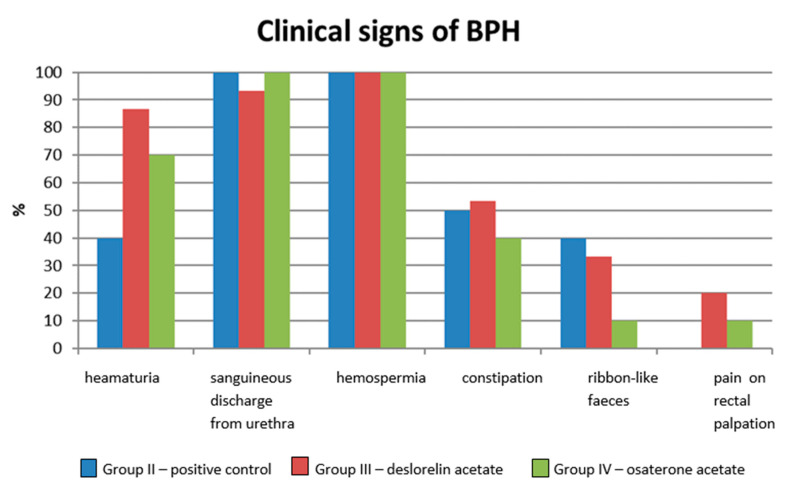
Clinical signs related to BPH in dogs included in Control Group II and Treated Groups III and IV, on Day 0 before any treatment.

**Figure 3 animals-10-01936-f003:**
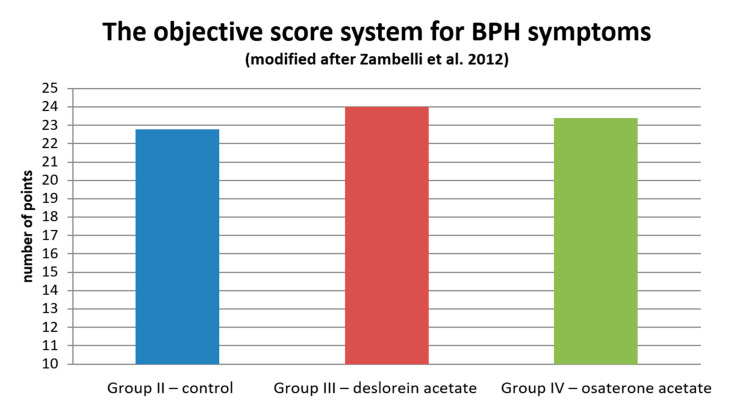
The severity and duration of BPH symptoms assessed using the objective scoring system modified after Zambelli et al. [20], on Day 0 before any treatment.

**Figure 4 animals-10-01936-f004:**
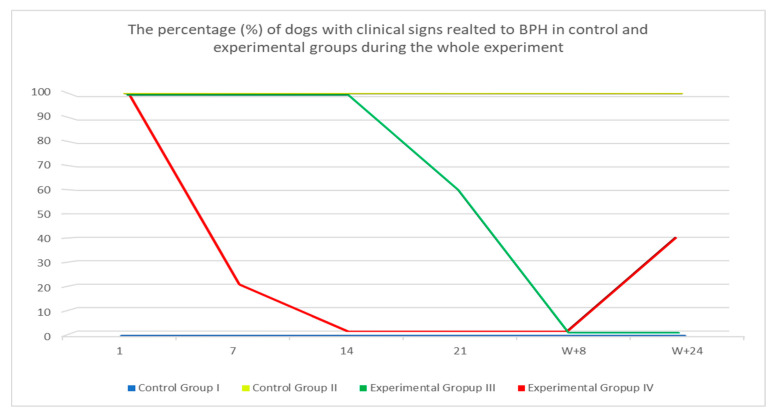
The percentage (%) of investigated dogs with clinical signs related to BPH Control Group I, II and Treated Groups III, IV during the whole observation period.

**Figure 5 animals-10-01936-f005:**
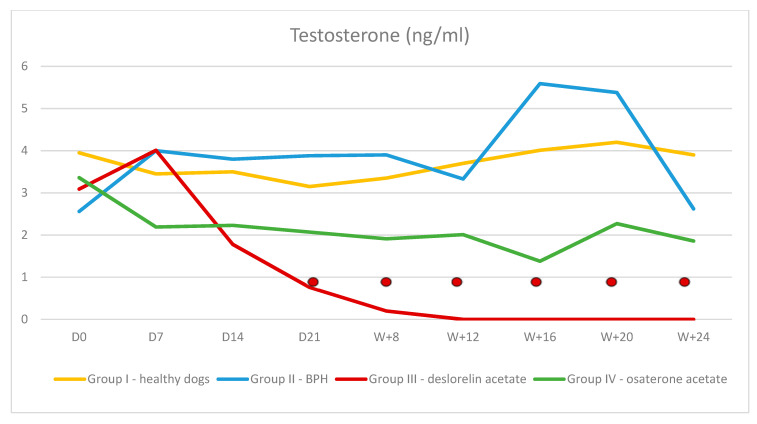
Concentration of testosterone in Control I and II and Treated III and IV Groups of dogs during 24 weeks of the trial (mean, SD). 
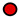
—indicates *p* < 0.01 compared to D0 in Group III.

**Figure 6 animals-10-01936-f006:**
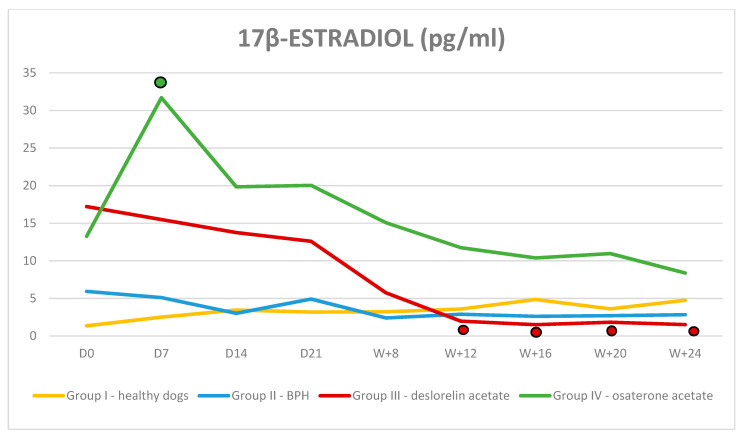
Concentration of 17β-estradiol in Control I and II and Treated III and IV Groups of dogs during 24 weeks of the trial (mean, SD). 
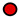

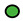
—indicates *p* < 0.01 compared to D0.

**Figure 7 animals-10-01936-f007:**
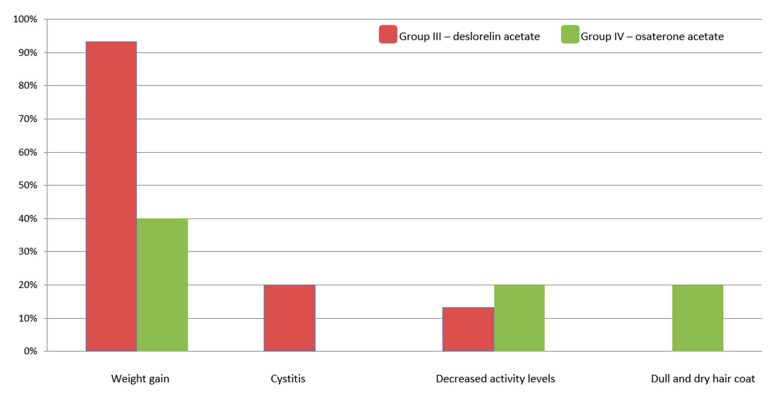
Treatment-related adverse effects observed in the treated Groups III and IV.

**Table 1 animals-10-01936-t001:** Number of dogs of particular breeds used in the study.

Breed and Number of Dogs Investigated	Group I	Group II	Group III	Group IV
Airedale Terrier				1
Alaskan Malamute		1		
American Staffordshire Terrier			1	2
Australian Cattle Dog	1	2		
Beagle	1		1	
Bedlington Terrier	1			
Bernese Mountain Dog			1	
Boxer			2	
Briard			1	
Dachshund			1	
German Shepherd				1
German Shorthaired Pointer				1
Irish Setter		1	1	1
Labrador Retriever			1	
Leonberger			1	1
Miniature Schnauzer			1	1
Pointer	1			
Polish Hound	2			
Shetland Sheepdog				1
Welsh Corgi Cardigan	2			
Yorkshire terrier		2		1
Cross Breed	2	4	4	
Total	10	10	15	10

**Table 2 animals-10-01936-t002:** The objective score system for benign prostate hyperplasia (BPH) signs grading, after Zambelli et al [20].

Anorexia	Weight Loss	Defecation	Dysuria	Urinary Incontinence	Urinary Leakage			Hematuria		
					Amount	Frequency	Duration	Amount	Frequency	Duration
1: absent	1: absent	1: normal	1: normal	1: absent	1: absent	1: absent	1: absent	1: absent	1: absent	1: absent
2: for 1 day	2: mild	2: regular with tenesmus	2: flow weaker or interrupted very	2: present	2: few drops	2: once in a year	2: one week	2: few drops	2: once in a year	2: one week
3: 1 to 7 days	3: moderate	3: regular with tenesmus	2: flow weaker or interrupted very		3: urine	3: 2–3 times per year	3: two week	3: pinkish drops	3: 2–3 times	3: two week
4: > 7 days	4: severe	4: absent for last a few days	4: urinary retention		4: copious	4: > 3 times per year	4: > 15 days	4: red urine	4: >3 times per year	4: > 15 days

**Table 3 animals-10-01936-t003:** Comparison of selected blood morphology and biochemistry parameters between control (I and II) and experimental (III and IV) groups in relation to day 0 (D0), week 8 (W + 8) and week 24 (W + 24) (mean, SD).

**Parameter**	**RBC (T/L)**				**HGB (mmol/L)**				**HCT (L/L)**				**MCV (f/L)**				**MCH (f/ML)**			
**Group**	I	II	III	IV	I	II	III	IV	I	II	III	IV	I	II	III	IV	I	II	III	IV
**D0**	6.67 ± 0.83	7.3 ± 0.5	7.61 ± 1.34	7.16 ± 0.92	9.68 ± 1.13	11.19 ± 2.19	9.99 ± 0.92	9.87 ± 1.34	0.49 ± 0.07	0.53 ± 0.03	0.51 ± 0.05	0.51 ± 0.07	73.1 ± 3.14	71.5 ± 2.84	73.01 ± 3.35	71.2 ± 4.02	1.45 ± 0.07	1.41 ± 0.05	1.45 ± 0.08	1.38 ± 0.09
**T + 8**	6.5 ± 0.48	7.13 ± 0.42	6.91 ± 0.54	6.13 ± 1.24	*9.51 ± 0.81*	*10.03 ± 0.44*	*9.91 ± 0.81*	*8.5 ± 1.68*	0.47 ± 0.05	0.51 ± 0.02	0.5 ± 0.04	0.43 ^&^ ± 0.09	72.67 ± 3.46	71.8 ± 1.55	72.87 ± 3.29	70 ± 4.85	1.46 ± 0.08	1.41 ± 0.04	1.42 ± 0.07	1.39 ± 0.14
**T + 24**	*6.57 ± 0.5*	*6.52 * ± 0.79*	*7.1 ± 0.56*	*7.16 ± 0.92*	*9.6 ± 1.01*	*9.4 * ± 1.14*	*10.66 ± 1.62*	*9.87 ± 1.34*	*0.48 ± 0.05*	*0.47 * ± 0.06*	*0.51 ± 0.05*	*0.51 ± 0.07*	72.56 ± 3.78	71.6 ± 1.51	72.33 ± 3.27	71.2 ± 4.02	1.46 ± 0.06	1.44 ± 0.03	1.44 ± 0.08	1.38 ± 0.09
**Parameter**	RBC (T/L)				HGB (mmol/L)				HCT (L/L)				MCV (f/L)				MCH (f/ML)			
**Group**	I	II	III	IV	I	II	III	IV	I	II	III	IV	I	II	III	IV	I	II	III	IV
**D0**	6.67 ± 0.83	7.3 ± 0.5	7.61 ± 1.34	7.16 ± 0.92	9.68 ± 1.13	11.19 ± 2.19	9.99 ± 0.92	9.87 ± 1.34	0.49 ± 0.07	0.53 ± 0.03	0.51 ± 0.05	0.51 ± 0.07	73.1 ± 3.14	71.5 ± 2.84	73.01 ± 3.35	71.2 ± 4.02	1.45 ± 0.07	1.41 ± 0.05	1.45 ± 0.08	1.38 ± 0.09
**T + 8**	6.5 ± 0.48	7.13 ± 0.42	6.91 ± 0.54	6.13 ± 1.24	*9.51 ± 0.81*	*10.03 ± 0.44*	*9.91 ± 0.81*	*8.5 ± 1.68*	0.47 ± 0.05	0.51 ± 0.02	0.5 ± 0.04	0.43 ^&^ ± 0.09	72.67 ± 3.46	71.8 ± 1.55	72.87 ± 3.29	70 ± 4.85	1.46 ± 0.08	1.41 ± 0.04	1.42 ± 0.07	1.39 ± 0.14
**T + 24**	*6.57 ± 0.5*	*6.52 * ± 0..79*	*7.1 ± 0.56*	*7.16 ± 0.92*	*9.6 ± 1.01*	*9.4 * ± 1.14*	*10.66 ± 1.62*	*9.87 ± 1.34*	*0.48 ± 0.05*	*0.47 * ± 0.06*	*0.51 ± 0.05*	*0.51 ± 0.07*	72.56 ± 3.78	71.6 ± 1.51	72.33 ± 3.27	71.2 ± 4.02	1.46 ± 0.06	1.44 ± 0.03	1.44 ± 0.08	1.38 ± 0.09
**Parameter**	**PLT (G/L)**				**WBC (G/L)**				**LYM (%)**				**MON (%)**				**GRA (%)**			
**Group**	I	II	III	IV	I	II	III	IV	I	II	III	IV	I	II	III	IV	I	II	III	IV
**D0**	286.3 ± 98.1	306.7 ± 100.88	261.13 ± 58.51	278.1 ± 125.71	8.89 ± 2.17	8.57 ± 1.89	7.61 ± 1.34	8.53 ± 2.4	14.23 ± 4.18	16.57 ± 3.37	14.31 ± 3.97	12.14 ± 2.91	4.82 ± 1.45	4.53 ± 0.84	5.28 ± 1.77	5.52 ± 2.3	80.95 ± 5.28	78.9 ± 3.81	80.41 ± 4.72	82.34 ± 4.9
**T + 8**	292 ± 112.71	281.3 ± 97.18	262.27 ± 61.99	313.5 ± 148.6	9.37 ± 2.57	9.03 ± 2.9	7.01 ± 1.09	6.92 ± 2.61	13.09 ± 3.51	18.49 ^&^ ± 2.49	15.13 ± 3.75	15.47 ± 3.02	4.3 ± 1.9	5.11 ± 1.21	5.13 ± 1.37	5.88 ± 1.87	82.61 ± 4.34	76.4 ± 3.32	79.74 ± 4.89	78.65 ± 3.91
**T + 24**	307.44 ± 88.53	269.1 ± 76.68	264.33 ± 82.14	278.1 ± 125.71	9.06 ± 2.1	9.8 ± 2.98	7.67 ± 1.41	8.53 ± 2.39	14.78 ± 5.11	16.12 ± 2.97	15.19 ± 4.21	12.14 ± 2.91	4.87 ± 1.53	4.62 ± 1.05	4.95 ± 1.23	5.52 ± 2.3	80.36 ± 6.08	79.26 ± 3.64	79.87 ± 5.25	82.34 ± 4.9
**Parameter**	**ALT (U/l)**				**AST (U/l)**				**ALP (U/l)**				**CREA (mg/dl)**				**UREA (mg/dl)**			
**Group**	I	II	III	IV	I	II	III	IV	I	II	III	IV	I	II	III	IV	I	II	III	IV
**D0**	53.94 ± 21.85	41.21 ± 19.08	57.94 ± 17.05	55.8 ± 40.17	31.15 ± 8.31	24.75 ± 8.21	29.89 ± 4.15	30.16 ± 7.65	23.42 ± 13.98	20.74 ± 3.77	36.42 ± 30.96	29.18 ± 15.89	0.93 ± 0.16	0.96 ± 0.27	0.91 ± 0.22	0.94 ± 0.19	33.71 ± 12.56	41.69 ± 10.01	36.65 ± 8.05	41.69 ± 27.55
**T + 8**	*55.32 ± 14.37*	*49.78 ± 33.44*	*58.59 ± 40.7*	*51.62 ± 25.27*	*27.46 ± 7.06*	*26.85 ± 5.64*	*31.81 ± 8.65*	*26.73 ± 3.2*	*19 ± 0*	*22.34 ± 8.92*	*28.75 ± 23.59*	*52.27 ± 93.17*	*0.89 ± 0.19*	*0.89 ± 0.21*	*1.01 ± 0.28*	*1.01 ± 0.34*	*39.78 ± 10.62*	*42.36 ± 21.48*	*45.22 * ± 10.43*	*43.61 ± 27.49*
**T + 24**	55.79 ± 20.66	49.31 ± 12.98	58.65 ± 47.33	43.22 ± 17.18	27.42 ± 5.63	31.29 ± 8.66	33.13 ± 9.77	29.63 ± 6	19 ± 0	28.13 ± 21.9	28.63 ± 22.99	42.62 ± 41.01	37.9 ± 0.17	0.94 ± 0.24	0.98 ± 0.3	1.03 ± 0.45	37.9 ± 12.65	49.7 ± 20.61	41.91 ± 11.59	39.65 ± 24.73

^&^*p* < 0.05 in relation to D0. * *p* < 0.01 in relation to D0. In italics—compared using Mann–Whitney test. Normal fonts—Student’s *t*-test.

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
