# Peer review of "Comparison of Clinical Effectiveness of Deslorelin Acetate and Osaterone Acetate in Dogs with Benign Prostatic Hyperplasia"

_animals, 2020, doi:10.3390/ani10101936_

Round 1

Reviewer 1 Report

Clinical effectiveness of deslorelin acetate and osaterone acetate in dogs with benign prostatic hyperplasia by NiżaÅ„ski W et al. is well written  manuscript containing relevant results concerning pharmacological treatment of enlarged prostate gland in dogs. In this MS effectiveness of potential therapeutic drug deslorelin acetate was compared with the approved one - osaterone acetate.

My concerns are as follow:

  1. The Authors have written that pharmacological treatment is considered in cases when fertility preservation or high testosterone level matters, but they did not present fertility assessment in DA treated dogs, as well as unchanged testosterone level in that group of treated dogs. My advice is to add comment in the Discussion and Conclusion to address that issue.
  2. It would be of value to add the results of prostate biopsy into the MS.
  3. Why the Authors did not add results of sperm analysis before and after the trial, in spite they had taken sperm samples to confirm BPH?
  4. I think that it would be better to show the flare-up effect of DA in Group III, e.g. by describing them because this effect was not justified by any data presented in the MS.
  5. Probably the Author may omit all three tables with hematological results or compress them to only one with data form starting point and the end point as there were no obvious effects of treatment on these parameters.
  6. Please try to more clearly present data in table2 marked as Figure 1 (for example change vertical table to the horizontal one to avoid splitting words)
  7. I would suggest to change X axis description in Fig. 3, 4, 8 because spaces and tabs are seen.
  8. I would suggest to use English version of cited books (No 17 and 30 in the References list) for the sake of international audience.
  9. Please check the language of MS, as I have found spelling error in the title of MS (effectivness)
  10. Is it correct with Journal requirements to put figure legends and title above the figure?

Author Response

Thank You for valuable comments. We hope our corrections meet all expectations and improved the manuscript

1. The Authors have written that pharmacological treatment is considered in cases when fertility preservation or high testosterone level matters, but they did not present fertility assessment in DA treated dogs, as well as unchanged testosterone level in that group of treated dogs. My advice is to add comment in the Discussion and Conclusion to address that issue.

The following sentence was added to the Discussion section:

Although, as found by other authors it does not impair the process of spermatogenesis [9, 13].

Line: 392 (in the corrected version)

This paper aimed to present the clinical changes related to the use of both medication. The data on fertility was analysed simultaneously, but due to the huge amount of results it was decided to address and discuss all fertility related issues in a separate paper.

2. It would be of value to add the results of prostate biopsy into the MS.

Authors are very sorry for using the term: prostate biopsy while there was FNA (fine needle aspirate) performed, which is a cytological assessment. The mistake was corrected throughout the text.

3. Why the Authors did not add results of sperm analysis before and after the trial, in spite they had taken sperm samples to confirm BPH?

As mentioned above the abundance of data made it impossible to address all the results in one paper. The authors elaborated the following manuscript presenting the fertility analysis, containing abundant number of results regarding detailed semen features and spem cells characteristic.

4. I think that it would be better to show the flare-up effect of DA in Group III, e.g. by describing them because this effect was not justified by any data presented in the MS.

The following sentence was added into Results section: the flare-up effect was noted (aggravation of clinical signs i.e. hematuria, sanguineous discharge from the prepuce, hemospermia, tenesmus or constipation)

Line: 217-218 (in the corrected version)

5. Probably the Author may omit all three tables with hematological results or compress them to only one with data form starting point and the end point as there were no obvious effects of treatment on these parameters.

Tables were compressed into one table, however authors decided to keep the previous time points to show that some changes were noted throughout the investigated period.

6. Please try to more clearly present data in table2 marked as Figure 1 (for example change vertical table to the horizontal one to avoid splitting words)

Figure 1 was changed to Table 2, Authors are very sorry for this mistake. In the submitted version the table was horizontal, it was changed into vertical by the editorial office.

7. I would suggest to change X axis description in Fig. 3, 4, 8 because spaces and tabs are seen.

These Figures were edited in such way by the Editorial Office, in original, submitted version spaces and tabs were not visible.

8. I would suggest to use English version of cited books (No 17 and 30 in the References list) for the sake of international audience.

Corrected

9. Please check the language of MS, as I have found spelling error in the title of MS (effectivness)

Corrected to ‘effectiveness’ Authors are sorry for this mistake.

10. Is it correct with Journal requirements to put figure legends and title above the figure?

The figures were edited in such way by the editorial system.

Reviewer 2 Report

I do think the study is well designed and the study methods are appropriate.

However there are some concerns

2 CLINICAL EFFECTIVNESS OF DESLORELIN

3 ACETATE AND OSATERONE ACETATE IN DOGS

4 WITH BENIGN PROSTATIC HYPERPLASIA

The title should be more clearer, as the drugs are not used together. And DA is compared to already approved OA.

12 Simple Summary: The article compares the treatment efficacy and side effects of two drugs used

Please replace side by adverse

29 clinical response, testosterone and estradiol levels, hematology, biochemistry and side effects

Please replace side by adverse

43 condition seen in male dogs.

Add intact

46 by type II 5α-reductase into dihydrotestosterone (DHT)

I think you should specify cell type if know like epithelial/or stromal cells

50 Despite that, at the beginning the condition is

51 often unnoticed, and the reason for consultation would be the reduced quality of life or

52 accompanying symptoms i.e. locomotor disorders, hyperthermia, unexplained deterioration of body

53 condition or GI symptoms (constipation, tenesmus or intermittent diarrhoea).

And 56 sometimes rectal obstruction [6]

Please rephrase. I do not understand ‘rectal obstruction’

63 e.g. due to a high anesthetic risk related to age

Why is it ‘high’ – reference?

69 and inhibiting the action of 5a-reductase.

Please stay consistent 5α

76 the initial flare-up effect upon the pituitary gland

Please rephrase ‘initial flare-up’

84 This article presents the results of a randomized clinical trial, designed to compare treatment

85 efficacy and therapeutic profiles of osaterone acetate or deslorelin acetate in male dogs..

Please change this according to the title. Remove the extra point.

119 Figure 1. The objective score system for BPH signs grading, after Zambelli et al.:

This is not a figure. This is terribly formatted.

09 analysis and serum hormonal analysis; FNA biopsy of the prostate (US-guided FNA) in order to

110 confirm BPH by cytological criteria [17, 18]

Results of the ultrasound, and the cytology of the prostate are not discussed. FNA cannot be a biopsy, it is just cytology

189 Straining during defecation, constipation, ribbon-like appearance of feces, or constipation were seen

Constipation is repeated

Figure 3. Clinical signs related to BPH in dogs included in Control Group II and Treated Groups III

196 and IV, on Day 0 before any treatment.

Key is very confusing

Figure 4. The severity and duration of BPH symptoms assessed using the objective scoring system

199 modified after Zambelli et al. [19], on Day 0 before any treatment.

Color should match the earlier figure 3 to make it less confusing

Figure 6. Concentration of testosterone in Control I and II and Treated III and IV Groups of dogs

249 during 24 weeks of the trial (mean, SD). – indicates p<0,01 compared to D0

Please do 1 graph vers 4 graphs. Testosterone spelling

Figure 7. Concentration of 17β-estradiol in Control I and II and Treated III and IV Groups of dogs

254 during 24 weeks of the trial (mean, SD). – indicates p<0,01 compared to D0.

Please do one graph versus 4 graphs

I do not see a need for the following tables; if it is for’ liver or kidney function nor bone marrow activity.’ As described in the discussion. These are not necessary

258 Table 2. Comparison of red blood cell parameters between control (I and II) and treated (III and IV)

259 groups in relation to D0 (D0, W+8, W+24) (mean, SD).

262 Table 3. Comparison of white blood cell and platelets count between control (I and II) and treated (III

263 and IV) groups in relation to D0 (D0, W+8, W+24) (mean, SD)

265 Table 4. Comparison of selected biochemistry parameters between control (I and II) and treated (III

266 and IV) groups in relation to D0 (D0, W+8, W+24) (mean, SD).

Figure 8. Treatment related side effects observed in the treated Groups III and IV.

Please stick to the same color codes

The purpose of this clinical trial was to evaluate the therapeutic efficacy and therapeutic profiles

299 of two medications used in BPH treatment for dogs: osaterone acetate (OA) and deslorelin acetate

300 (DA).

Please change with the title, and the objective

The only published trials presenting the use of DA in dogs was performed in clinically

332 asymptomatic dogs, in which the prostate gland showed signs of enlargement only on US and

333 histological examinations

Histology were cytology were not discussed

397 anal glands adenocarcinoma.??

This needs a reference

406 combined therapy might be considered, when DA implant is placed 1 – 2 weeks after OA

407 administration to achive better clinical response due to the avoidance of the draw backs of both

408 medications.

This conclusion is a stretch. All the study can say is that DA is better or just equal to OA. A subsequent study may look into this hypothesis

Surprised in the screening that no dogs with neoplasia was found - discuss

Why does the DA group have 15 versus all the other groups - this would go along with recommendation to change the title and objective 

Author Response

Thank You for valuable comments and suggestions. We did our best to address all the comments. We hope, that manusrcipt has been therefore improved.

The title should be more clearer, as the drugs are not used together. And DA is compared to already approved OA.

 The title was changed into: Comparison of clinical effectiveness of deslorelin acetate and osaterone acetate in dogs with benign prostatic hyperplasia

12 Simple Summary: The article compares the treatment efficacy and side effects of two drugs used

Please replace side by adverse

Changed according to reviewer suggestions here and throughout the text

29 clinical response, testosterone and estradiol levels, hematology, biochemistry and side effects

Please replace side by adverse

Changed according to reviewer suggestions here and throughout the text

43 condition seen in male dogs.

Add intact

Added

46 by type II 5α-reductase into dihydrotestosterone (DHT)

I think you should specify cell type if know like epithelial/or stromal cells

It was changed follows: and converted by type II 5α-reductase into dihydrotestosterone (DHT) in prostate, seminal vesicles, epididymies, skin, liver and brain.

50 Despite that, at the beginning the condition is

51 often unnoticed, and the reason for consultation would be the reduced quality of life or

52 accompanying symptoms i.e. locomotor disorders, hyperthermia, unexplained deterioration of body

53 condition or GI symptoms (constipation, tenesmus or intermittent diarrhoea).

And 56 sometimes rectal obstruction [6]

Please rephrase. I do not understand ‘rectal obstruction’

The phrase was removed, authors agree it was not the best worded

63 e.g. due to a high anesthetic risk related to age

Why is it ‘high’ – reference?

It was changed into ‘higher’ and reference was added

69 and inhibiting the action of 5a-reductase.

Please stay consistent 5α

Authors are sorry for this mistake and it was corrected

76 the initial flare-up effect upon the pituitary gland

Please rephrase ‘initial flare-up’

It was changed into: after the initial rise of pituitary gland hormones

84 This article presents the results of a randomized clinical trial, designed to compare treatment

85 efficacy and therapeutic profiles of osaterone acetate or deslorelin acetate in male dogs..

Please change this according to the title. Remove the extra point.

It was changed into: This article presents the comparison of the treatment of osaterone acetate or deslorelin acetate in male dogs suffering from benign prostate hyperplasia.

119 Figure 1. The objective score system for BPH signs grading, after Zambelli et al.:

This is not a figure. This is terribly formatted.

 The title was changed for Table, authors are very sorry for such mistake. However, the table formatting was done by the editorial office, In original, submitted version it was horizontal and easier to read.

09 analysis and serum hormonal analysis; FNA biopsy of the prostate (US-guided FNA) in order to

110 confirm BPH by cytological criteria [17, 18]

Results of the ultrasound, and the cytology of the prostate are not discussed. FNA cannot be a biopsy, it is just cytology

The word ‘biopsy’ was removed, authors are very sorry for this mistake. The FNA in this study was performed only to confirm BPH by cytological criteria. If the FNA results suggested any other disorder such dog was excluded from this study. The US results were also collected in this study, but these data, in authors opinion, requires separate analysis and publication.

189 Straining during defecation, constipation, ribbon-like appearance of feces, or constipation were seen

Constipation is repeated

removed

Figure 3. Clinical signs related to BPH in dogs included in Control Group II and Treated Groups III

196 and IV, on Day 0 before any treatment.

Key is very confusing

The Key was formatted by editorial office in such way, in submitted version it was more clear.

Figure 4. The severity and duration of BPH symptoms assessed using the objective scoring system

199 modified after Zambelli et al. [19], on Day 0 before any treatment.

Color should match the earlier figure 3 to make it less confusing

 Colours were adjusted to match the previous table.

Figure 6. Concentration of testosterone in Control I and II and Treated III and IV Groups of dogs

249 during 24 weeks of the trial (mean, SD). – indicates p<0,01 compared to D0

Please do 1 graph vers 4 graphs. Testosterone spelling

Testosterone spelling was corrected.

Graphs were merged into 1 graph.

Figure 7. Concentration of 17β-estradiol in Control I and II and Treated III and IV Groups of dogs

254 during 24 weeks of the trial (mean, SD). – indicates p<0,01 compared to D0.

Please do one graph versus 4 graphs

 Graphs were merged into 1 graph.

I do not see a need for the following tables; if it is for’ liver or kidney function nor bone marrow activity.’ As described in the discussion. These are not necessary

258 Table 2. Comparison of red blood cell parameters between control (I and II) and treated (III and IV)

259 groups in relation to D0 (D0, W+8, W+24) (mean, SD).

262 Table 3. Comparison of white blood cell and platelets count between control (I and II) and treated (III

263 and IV) groups in relation to D0 (D0, W+8, W+24) (mean, SD)

265 Table 4. Comparison of selected biochemistry parameters between control (I and II) and treated (III

266 and IV) groups in relation to D0 (D0, W+8, W+24) (mean, SD).

The tables were compressed them into 1 table, as this was suggested by the other reviewer.

Figure 8. Treatment related side effects observed in the treated Groups III and IV.

Please stick to the same color codes

Same colour codes were used.

The purpose of this clinical trial was to evaluate the therapeutic efficacy and therapeutic profiles

299 of two medications used in BPH treatment for dogs: osaterone acetate (OA) and deslorelin acetate

300 (DA).

Please change with the title, and the objective

It was changed into: The purpose of this clinical study was to compare the therapeutic efficacy

of two medications used in BPH treatment for dogs: osaterone acetate (OA) and deslorelin acetate (DA).

The only published trials presenting the use of DA in dogs was performed in clinically

332 asymptomatic dogs, in which the prostate gland showed signs of enlargement only on US and

333 histological examinations

Histology were cytology were not discussed

The sentence quoted by the reviewer is the only available publications on DA use in dogs, hence authors decided to mentioned it. However, the reviewer is right, in the presented study the histology was not performed. Authors added the word ‘clinical’ before the word ‘symptoms’ in the following sentence concerning our study. Authors hope it clears the difference between published and our data. 

397 anal glands adenocarcinoma.??

This needs a reference

Added

406 combined therapy might be considered, when DA implant is placed 1 – 2 weeks after OA

407 administration to achive better clinical response due to the avoidance of the draw backs of both

408 medications.

This conclusion is a stretch. All the study can say is that DA is better or just equal to OA. A subsequent study may look into this hypothesis

 Authors agree with the reviewer and changed the sentence into:

Based on the obtained results in some cases it might be justified to consider the combined therapy with both medications. The OA to be given first and followed by DA 1 – 2 weeks later. Such approach should lead to a better clinical efficacy, reducing the adverse effects of both medications, i.e. exacerbation of clinical symptoms seen initially with DA and the limited time of clinical improvement seen with the use of OA.

Surprised in the screening that no dogs with neoplasia was found - discuss

As mentioned above, for the study were enrolled only dogs with BPH confirmed by FNA. Dogs with other FNA results were excluded  from this study. 

Why does the DA group have 15 versus all the other groups - this would go along with recommendation to change the title and objective 

The DA is a quite new medication, there are not many published data on its use and effect in dogs. Whereas there are substantial data published on the use of OA in dogs. That is why authors decided to analyse more dogs receiving DA. Authors hoped this approach would allow to obtain more accurate results for this group of dogs.

Reviewer 3 Report

The rational behind the experiment was clear and straight forward. The manuscript is almost well written

While many different sources are used to set up the study in the introduction, little previous evidence is stated. The introduction is thus short and poorly sets up the rationale for the study. More attention to how this study fits into previous work in BPH and inflammation should be added to improve this section. Please refer to doi: 10.1016/j.taap.2017.06.005

There are some minor grammar issues that should be fixed in order to aid the accessibility of the results to the reader.

Author Response

Thank You for comments. We hope we addressed them correctly. 

While many different sources are used to set up the study in the introduction, little previous evidence is stated. The introduction is thus short and poorly sets up the rationale for the study. More attention to how this study fits into previous work in BPH and inflammation should be added to improve this section. Please refer to doi: 10.1016/j.taap.2017.06.005

The Introduction was improved and reference added.

There are some minor grammar issues that should be fixed in order to aid the accessibility of the results to the reader.

English language was proof read

Round 2

Reviewer 1 Report

I have seen that all my concerns were answered by the Authors in the revised version of the MS. It means that I don't have any other comments to the MS.

Author Response

Authors would like to thank the reviewer for the time and effort which definitely helped to improve our MS.

Reviewer 2 Report

70 progestogens to experimental therapies utilizing micronized endogenous fatty acid amide and
71 resveratrol precursor [2, 5, 10, 11]. However,

should be corrected as amides and precursor

355 subtle, the antiandrogenic effect of OA cannot be fully excluded [24]. Although, as found by other
356 authors it does not impair the process of spermatogenesis [9, 14]

Please rephrase.

408 use of OA.

I think before such a recommendation, additional studies would be warranted. 

Author Response

Authors would like to thank the reviewer for comments and suggestions which help to improve the MS. All suggested changes have been addressed as follows:

70 progestogens to experimental therapies utilizing micronized endogenous fatty acid amide and
71 resveratrol precursor [2, 5, 10, 11]. However, should be corrected as amides and precursor

Corrected

355 subtle, the antiandrogenic effect of OA cannot be fully excluded [24]. Although, as found by other authors it does not impair the process of spermatogenesis [9, 14]

Please rephrase.

It was changed into: Hence, although subtle, the antiandrogenic effect of OA cannot be fully excluded [24], but as found by other authors it did not impair the process of spermatogenesis [9, 14].

408 use of OA.

I think before such a recommendation, additional studies would be warranted. 

The sentence was changed as follows: Additional studies are needed to check whether the combined therapy using OA and DA would lead to a better clinical efficacy, reducing the adverse effects of both medications, i.e. exacerbation of clinical symptoms seen initially with DA and the limited time of clinical improvement seen with the use of OA.